# Accelerating physics-informed neural fields for fast CT perfusion analysis in acute ischemic stroke

**Lucas de Vries**[1,2,3]                                LUCAS.DEVRIES@AMSTERDAMUMC.NL

**Rudolf L. M. van Herten**[2,3]                    R.L.M.VANHERTEN@AMSTERDAMUMC.NL

**Jan W. Hoving**[1]                                          J.W.HOVING@AMSTERDAMUMC.NL

**Ivana Išgum**[1,2,3]                                       I.ISGUM@AMSTERDAMUMC.NL

**Bart J. Emmer**[1]                                        B.J.EMMER@AMSTERDAMUMC.NL

**Charles B. Majoie**[1]                                  C.B.MAJOIE@AMSTERDAMUMC.NL

**Henk A. Marquering**[1,2]                         H.A.MARQUERING@AMSTERDAMUMC.NL

**Efstratios Gavves**[3]                                  E.GAVVES@UVA.NL

[1] *Amsterdam UMC location University of Amsterdam, Radiology and Nuclear Medicine, Meibergdreef 9, Amsterdam, 1105 AZ, The Netherlands*

[2] *Amsterdam UMC location University of Amsterdam, Biomedical Engineering and Physics, Meibergdreef 9, Amsterdam, 1105 AZ, The Netherlands*

[3] *Informatics Institute, University of Amsterdam, Amsterdam, The Netherlands*

**Editors:** Accepted for publication at MIDL 2024

## Abstract

Spatio-temporal perfusion physics-informed neural networks were introduced as a new method (SPPINN) for CT perfusion (CTP) analysis in acute ischemic stroke. SPPINN leverages physics-informed learning and neural fields to perform a robust analysis of noisy CTP data. However, SPPINN faces limitations that hinder its application in practice, namely its implementation as a slice-based (2D) method, lengthy computation times, and the lack of infarct core segmentation. To address these challenges, we introduce a new approach to accelerate physics-informed neural fields for fast, volume-based (3D), CTP analysis including infarct core segmentation: ReSPPINN. To accommodate 3D data while simultaneously reducing computation times, we integrate efficient coordinate encodings. Furthermore, to ensure even faster model convergence, we use a meta-learning strategy. In addition, we also segment the infarct core. We employ acute MRI reference standard infarct core segmentations to evaluate ReSPPINN and we compare the performance with two commercial software packages. We show that meta-learning allows for full-volume perfusion map generation in 1.2 minutes without comprising quality, compared to over 40 minutes required by SPPINN. Moreover, ReSPPINN's infarct core segmentation outperforms commercial software.

**Keywords:** neural fields, CT perfusion, physics-informed, acute ischemic stroke

## 1. Introduction

CT perfusion (CTP) imaging is often part of the imaging work-up of patients suffering acute ischemic stroke for treatment decision support. CTP is the sequential acquisition of CT after contrast agent administration. These images are subsequently processed for the generation of so-called *perfusion maps* (Konstas et al., 2009). These maps depict, for example, the cerebral blood flow and time-to-maximum contrast attenuation, which are crucial

in estimating the infarct core (irreversibly damaged tissue) and penumbra (hypoperfused but salvageable tissue). Typical commercial CTP software thresholds the perfusion maps to determine these two regions (Demeestere et al., 2020).

In recent years, many deep learning-based studies focused on performing infarct core segmentation from commercial vendor CTP perfusion maps (Chen et al., 2020; Clèrigues et al., 2019; Abulnaga and Rubin, 2019), infarct core segmentation directly from CTP source data (Bertels et al., 2019; Robben et al., 2020; De Vries et al., 2023a), or a combination of both (Wang et al., 2020). On the other hand, De Vries et al. (2023b) used deep learning for the generation of perfusion maps and introduced SPPINN, a novel approach to CTP analysis using spatio-temporal physics-informed neural networks, which showed improved accuracy for estimating perfusion parameters particularly when data are noisy. SPPINN infers the perfusion parameters by learning coordinate-based neural networks, or *neural fields*, that represent the perfusion parameters and observed data, guided by a loss function formulated as the residual of a differential equation corresponding to the dynamics of CTP.

SPPINN's application in clinical settings is considerably limited due to (i) the method operating on 2D axial slices rather than the full 3D volume, (ii) the computation time being too long for clinical use, and (iii) the lack of infarct core segmentation. To address these limitations, we introduce ReSPPINN for fast, volume-based (3D), CTP analysis with physics-informed neural fields including infarct core segmentation. To adapt from a slice-based to a volume-based method and at the same time reduce convergence time, we use coordinate encodings. Additionally, we learn network initializations through meta-learning to further reduce computation time. To expand the potential of ReSPPINN in a clinical setting, we incorporate infarct core segmentation. We evaluate our method on two levels. Initially, we assess the segmentation and detection performance of ReSPPINN against that of two commercial CTP software packages. Subsequently, we examine if the perfusion maps generated by ReSPPINN maintain the same level of accuracy for infarct core segmentation as perfusion maps which were not subject to acceleration. We achieve this by comparing its derived infarct core segmentation to acute MRI reference standard infarct core segmentations.

## 2. Method

In the following, we introduce ReSPPINN. Figure 1 presents an overview of our method.

### 2.1. Physics-informed neural fields for CTP analysis

In CTP source data, the arterial input function $C_{\text{AIF}}(t)$ is the contrast attenuation curve in one of the main supplying arteries, and $C_{\text{TAC},v}(t)$ is the tissue contrast attenuation in voxel $v$. The approach we follow to obtain the perfusion parameters from CTP source data is to infer the parameters of the following differential equation (Bennink et al., 2016):

$$\frac{dC_{\text{TAC},v}(t)}{dt} = \text{CBF}_v \cdot [C_{\text{AIF}}(t - t_{d,v}) - C_{\text{AIF}}(t - t_{d,v} - \text{MTT}_v)], \tag{1}$$

for each spatial voxel $v$. The cerebral blood flow ($\text{CBF}_v$), delay ($t_{d,v}$), and mean transit time ($\text{MTT}_v$) are the parameters to be inferred for voxel $v$. The cerebral blood volume is $\text{CBV}_v = \text{CBF}_v \times \text{MTT}_v$ and time-to-maximum results from $\text{Tmax}_v = t_d + \frac{1}{2}\text{MTT}_v$. To

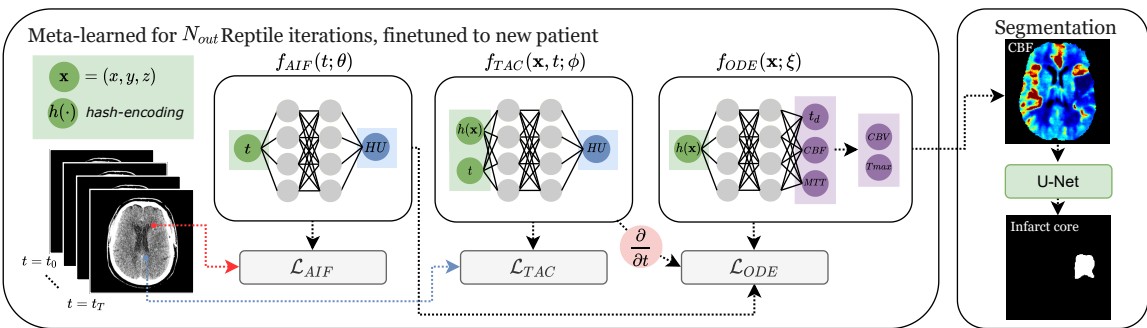

Figure 1: ResPPINN learns neural fields $f_{\text{TAC}}(\mathbf{x}, t; \phi)$ and $f_{\text{AIF}}(t; \theta)$ of the observed tissue attenuation and arterial input data, and $f_{\text{ODE}}(\mathbf{x}; \xi)$ for the perfusion parameters. With meta-learning, we learn neural field initializations and at test time we tune them to new patient data. We use the CBF map for infarct core segmentation.

solve for the perfusion parameters of voxel $v$, we derive the residual form of Equation (1):

$$r(t) = \frac{dC_{\text{TAC},v}(t)}{dt} - \text{CBF}_v \cdot [C_{\text{AIF}}(t - t_{d,v}) - C_{\text{AIF}}(t - t_{d,v} - \text{MTT}_v)], \tag{2}$$

We can define an objective function, e.g. $\mathcal{L} = r(t)^2$, and use an optimization method to minimize $\mathcal{L}$ and infer the parameters of the differential equation. This is not straightforward, however, since the temporal data for the tissue attenuation and arterial input function have a low temporal resolution (often 1-2 seconds), the attenuation curves are noisy, and $\frac{dC_{\text{TAC},v}(t)}{dt}$ is not well-defined. We approach this problem by using neural fields and physics-informed learning (Raissi et al., 2019). A field $f : \mathbb{R}^d \to \mathbb{R}^n$ is a scalar ($n = 1$) or vector ($n > 1$) quantity defined over the spatial, temporal, or spatio-temporal domain ($d = 1, 2, 3, 4$), i.e. the Hounsfield units (the quantity) in a CT scan (the domain). In the case of CTP, the observed data are discretely sampled at coordinates $\mathbf{x} = (x, y, z)$ on the voxel grid and $t \in [t_0, ..., t_T]$ in the temporal domain $T$. Using fields we can rewrite Equation (2):

$$r(\mathbf{x}, t) = \frac{\partial C_{\text{TAC}}(\mathbf{x}, t)}{\partial t} - \text{CBF}(\mathbf{x}) \cdot [C_{\text{AIF}}(t - t_d(\mathbf{x})) - C_{\text{AIF}}(t - t_d(\mathbf{x}) - \text{MTT}(\mathbf{x}))], \tag{3}$$

A *neural* field is a neural network $f_\theta$ that parameterizes a field $f$ (Xie et al., 2022). We learn neural fields of the arterial input function data: $f_{\text{AIF}}(t)$, and tissue attenuation data: $f_{\text{TAC}}(\mathbf{x}, t)$, but also the perfusion parameters: $f_{\text{CBF}}(\mathbf{x})$, $f_{\text{MTT}}(\mathbf{x})$, $f_{t_d}(\mathbf{x})$. Since neural fields are fully differentiable, we can compute the continuous derivative $\frac{\partial C_{\text{TAC}}(\mathbf{x},t)}{\partial t}$. This allows us to rewrite Equation (3) with neural fields:

$$r(\mathbf{x}, t) = \frac{\partial f_{\text{TAC}}(\mathbf{x}, t)}{\partial t} - f_{\text{CBF}}(\mathbf{x}) \cdot [f_{\text{AIF}}(t - f_{t_d}(\mathbf{x})) - f_{\text{AIF}}(t - f_{t_d}(\mathbf{x}) - f_{\text{MTT}}(\mathbf{x}))]. \tag{4}$$

In the same fashion as Equation (2), we can define an objective function or *physics-informed* loss function, e.g. $\mathcal{L}_{ODE} = r(\mathbf{x}, t)^2$, and use stochastic gradient descent to minimize the loss and train the neural fields for the perfusion parameters. After training, we sample the perfusion parameter neural fields at the spatial domain to obtain the perfusion maps.

**Neural field definitions and training** $f_{\text{AIF}}(t; \theta)$: $\mathbb{R} \to \mathbb{R}$ and $f_{\text{TAC}}(\mathbf{x}, t; \phi)$: $\mathbb{R}^4 \to \mathbb{R}$ are parameterized with sets of parameters $\theta$ and $\phi$. In practice, we estimate the perfusion parameters with a single neural field $f_{\text{ODE}}(\mathbf{x}; \xi)$: $\mathbb{R}^3 \to \mathbb{R}^3$ that is only a function of the spatial coordinates. Hence, the parameters of $f_{\text{CBF}}(\mathbf{x})$, $f_{\text{MTT}}(\mathbf{x})$, $f_{\text{T}_d}(\mathbf{x})$ are shared. We use mini-batches $(\mathbf{x}, t)$ to optimize $f_{\text{AIF}}(t; \theta)$ and $f_{\text{TAC}}(\mathbf{x}, t; \phi)$ with norm-based loss functions $\mathcal{L}_{AIF}$ and $\mathcal{L}_{TAC}$, supervised with the observed data. We sample sets of continuous collocation points $(\mathbf{x}, \tau)$ with $\tau \in T_c$ in the same range as $T$ to minimize $\mathcal{L}_{ODE}$ and to ensure smooth derivatives between time points. The total loss is:

$$\mathcal{L} = \mathcal{L}_{AIF} + \mathcal{L}_{TAC} + \mathcal{L}_{ODE}. \tag{5}$$

For the specific implementation details and loss functions, we refer to Appendix A.

## 2.2. Coordinate encoding

Tancik et al. (2020) showed that simply using coordinates as input to the neural fields limits the capacity of neural fields to fit high-frequency details. This complexity to fit details increases with the dimensionality of the problem, particularly causing problems for slice-based SPPINN to be a full 3D(+T) approach. Tancik et al. (2020), therefore, proposed encoding the coordinates into a higher dimensional space to accelerate convergence. We follow the multi-resolution hash-encodings, in short, *hash-encodings* or $h(\cdot)$, proposed by (Müller et al., 2022) to encode our spatial coordinates into an efficient higher dimensional space. Hash-encodings define $L$ multi-resolution grids over the input domain with $d$ learnable weights at each grid point at each resolution. For a coordinate $\mathbf{x}$, it determines the closest grid points per resolution and linearly interpolates the weights at these grid points to obtain embedding $e \in \mathbb{R}^{Ld}$. We share the encoding layer between $f_{\text{ODE}}(\mathbf{x}; \xi)$ and $f_{\text{TAC}}(\mathbf{x}, t; \phi)$. For $f_{\text{AIF}}(t; \theta)$ we do not require encoding, since approximating the one-dimensional $C_{\text{AIF}}(t)$ is already efficient without hashing. The hash-encoding $h(\cdot)$ lacks global differentiability due to its discontinuities at hash grid boundaries, and with the discontinuous nature of the derivative in its linear interpolation. Therefore, to keep the temporal derivatives well-defined, we only encode spatial coordinates and not the temporal coordinates. This hash-encoding allows ReSPPINN to use small architectures (3 layers, 16 neurons) for 3D+T and 3D neural fields.

## 2.3. Learning initializations

Learning neural fields from random network initializations is inefficient. Tancik et al. (2021) showed that neural fields can be trained with considerably fewer steps when the initialization facilitates fast convergence. We use the *Reptile* meta-learning algorithm (Nichol et al., 2018) to learn an optimal initialization for $f_{\text{AIF}}(t; \theta)$, $f_{\text{TAC}}(\mathbf{x}, t; \phi)$, and $f_{\text{ODE}}(\mathbf{x}; \xi)$ using training data. Let us consider the neural field $f_{\text{TAC}}(\mathbf{x}, t; \phi)$ with parameters $\phi$. Reptile meta-learning consists of an outer and an inner loop, as described in Algorithm 1. In the outer loop, we learn the initialization of the neural fields in $N_{out}$ iterations. In the inner loop, we optimize the neural field $f_{\text{TAC}}(\mathbf{x}, t; \phi)$ for an instance in the training data for $N_{in} > 1$ iterations starting from the current initialization $\phi$ and obtain $\phi^*$. We set the difference between the parameters $\phi$ and $\phi^*$, scaled by $\epsilon$, as the gradient for the neural field. After each inner loop, we run gradient descent to update the neural field parameters. We select the network obtained after $N_{out}$ iterations to use for inference. We empirically set $N_{out} = 7500$ in our

experiment. Without meta-learned initialization, ReSPPINN empirically shows convergence after 5000 iterations. We, therefore, set the $N_{in} = 500$ to achieve a factor 10 speed-up.

---

**Algorithm 1:** Reptile meta-learning for ReSPPINN's $f_{\text{TAC}}(\mathbf{x}, t; \phi)$.

**Data:** Initialize $\phi$, the initial parameter vector for neural field $f$

**for** *iteration* $1, 2, 3, \ldots, N_{out}$ **do**
  Randomly sample a patient $P$
  Perform $N_{in} > 1$ steps on patient $P$, starting with parameters $\phi$, resulting in $\phi^*$
  Update: $\phi \leftarrow \phi + \epsilon(\phi^* - \phi)$
**end**
**return** $\phi$

---

### 2.4. Infarct core segmentation

Consistent with the methodologies of commercial CTP software, we use the CBF map for infarct core segmentation. Specifically, we calculate a relative CBF map by scaling it to the median CBF value of the healthy hemisphere. This relative map is the input to a U-Net (Ronneberger et al., 2015) and we use infarct segmentations from co-registered reference standard acute MRI for supervision. Appendix A presents more implementation details.

### 2.5. Baseline

We compare ReSPPINN to baseline SPPINN. We re-implemented SPPINN to align with our new approach for a fair comparison. SPPINN has two-dimensional inputs, no coordinate encoding scheme, no meta-learned initializations, and larger network architectures for $f_{\text{TAC}}(\mathbf{x}, t; \phi)$ (3 layers, 128 neurons) and $f_{\text{ODE}}(\mathbf{x}; \xi)$ (3 layers, 64 neurons). For a review of the vanilla SPPINN implementation and its quantitative performance, we refer to De Vries et al. (2023b).

### 2.6. Datasets

We use data from the CLEOPATRA health care evaluation study (Koopman et al., 2022). We included 898 patients who received CTP at baseline and for which the CTP scan was processed with CTP software StrokeViewer (version 3.2.11; Nicolab, Amsterdam, The Netherlands). For training the infarct core segmentation model, we allocated 15 patients who also underwent Diffusion-Weighted Imaging (DWI) MRI at baseline, only including imaging with an interval between CTP and DWI < 4.5 hours to limit the effect of infarct growth (Bala et al., 2021). The median (IQR) interval was 56 (41 − 70) minutes. Using a semi-automated method (Tolhuisen et al., 2022; Kamnitsas et al., 2017), we obtained the ground truth infarct core segmentations after co-registration to the CTP. We manually corrected the results as necessary. Those 15 patients also had results from commercial software Syngo.via CT Neuro Perfusion (version VB40; Siemens Healthcare, Erlangen, Germany) available. In our analysis, we use the CTP source data pre-processing (motion reduction, smoothing) and AIF from StrokeViewer. We aligned all scans to a standard coordinate frame of size $256 \times 256 \times 32$ with spacing 0.91 mm $\times$ 0.91 mm $\times$ 5.00 mm to ensure that the midline was properly centered.

## 3. Experiments

**Effectiveness of hash-encodings**   Preliminary experiments showed that using full CTP volumes causes problems in fitting the high-frequency details with SPPINN. We, therefore, investigate ReSPPINN's convergence speed with hash-encodings versus slice-based SPPINN. For comparison, we use ReSPPINN without learned initialization (ReSPPINN-no-init). We train volume-based ReSPPINN for 5000 iterations and use 5000 and 10000 iterations per slice for SPPINN. We compare $\mathcal{L}_{TAC}$ for ReSPPINN to the average loss over all slices for SPPINN.

**Accelerating convergence speed with Reptile meta-learning**   We compare $\mathcal{L}_{TAC}$ and $\mathcal{L}_{ODE}$ for ReSPPINN trained for 5000 iterations without meta-learned initialization (ReSPPINN-no-init@5000) and ReSPPINN@500, trained with only 10% of the iterations with initialization. We exclude $\mathcal{L}_{AIF}$ from evaluation as $f_{\mathrm{AIF}}(t;\theta)$ fits $C_{\mathrm{AIF}}(t)$ fast regardless of initialization. Furthermore, we investigate the computation time gain achieved by Reptile meta-learning and compare the total computation time to baseline SPPINN.

**Infarct core segmentation**   We use the CBF map as input to a segmentation model (Section 2.4). We train the same model for both ReSPPINN@500 and ReSPPINN-no-init@5000 CBF maps to investigate if the model with ReSPPINN@500 perfusion maps achieves similar performance for the downstream segmentation task, compared to using ReSPPINN-no-init@5000 maps. We train and evaluate through leave-one-out cross-validation on all patients (14 training, 1 test). We use the first fold to define the hyperparameters and exclude this fold from all evaluations. We measure the average Dice score, mean volumetric difference (MVD), and absolute volumetric difference (AVD), between the reference and automatic segmentations, and the false negative rate (FNR) for infarct detection. We compare ReSPPINN's infarct core segmentation results with those from two commercial vendors.

## 4. Results

**Effectiveness of hash-encodings**   Figure 2 (left) shows the tissue attenuation loss for one patient for SPPINN (in grey) and ReSPPINN-no-init (in blue). The hash-encodings allow ReSPPINN-no-init to fit the full-volume tissue attenuation data in 5000 iterations. Unlike the proposed method, SPPINN is unable to fit the high-frequency data in 5000 iterations per slice. The obtained data representation is less detailed which causes too smooth perfusion maps. Training SPPINN for more iterations will further reduce the loss but also increase the computation time, which is undesirable. Optimization differences between SPPINN@5000 and @10000 stem from our iteration-based learning rate scheduler. Figure 3 shows that SPPINN maps are less detailed compared to ReSPPINN, even after 10000 iterations.

**Accelerating convergence speed with meta-learning**   Figure 2 shows $\mathcal{L}_{TAC}$ (left) and $\mathcal{L}_{ODE}$ (center) for one patient for ReSPPINN-no-init@5000 (in blue) and the proposed ReSPPINN@500 (in red). We observe rapid and stable convergence for ReSPPINN@500, for both $\mathcal{L}_{TAC}$ and $\mathcal{L}_{ODE}$. Figure 2 shows the full-volume computation time (right) on an Nvidia V100 GPU. Training ReSPPINN-no-init@5000 until convergence takes approximately 12 minutes. Meta-learning allows for fast convergence in 1.2 minutes on average. SPPINN processes each slice in 1.7 minutes for 5000 iterations and 3.5 minutes for 10000 iterations, culminating in full-volume computation times of 40-50 and 60-100 minutes, respectively.

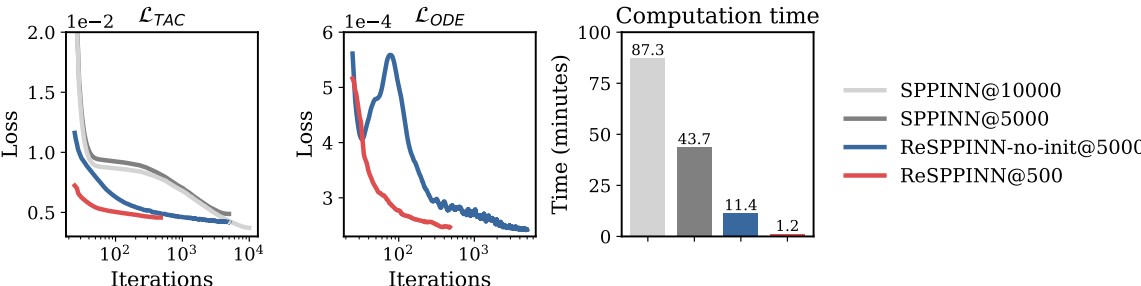

Figure 2: Loss curves and computation time for SPPINN and ReSPPINN. The losses are for a single patient and the computation times are averages over the validation set.

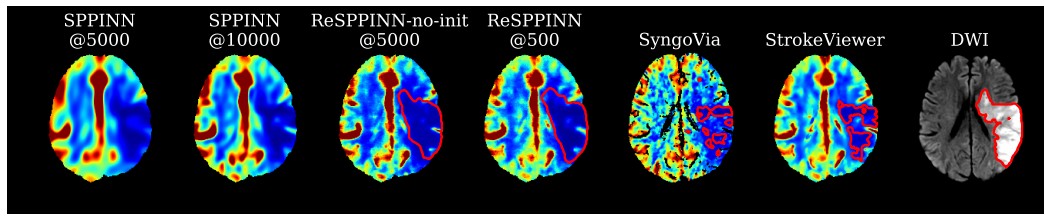

Figure 3: The CBF perfusion maps and DWI for one patient. The infarct core segmented by each method and the DWI reference segmentation are outlined in red.

**Infarct core segmentation**   Figure 3 shows ReSPPINN and commercial CBF maps for one patient, including the infarct core segmentations (in red) for these methods, and the DWI reference. The ReSPPINN CBF map shows a low CBF region at the location of the infarct. The visual differences between ReSPPINN-no-init@5000 and ReSPPINN@500 are marginal. The CBF map generated by Syngo.via shows irregular patterns and noticeable visual differences, primarily due to the exclusion of vessels. On the other hand, StrokeViewer generates results that are more similar to our method, but the perfusion map appears somewhat smoother and displays slightly elevated CBF within the infarcted area compared to ReSPPINN. By visual inspection, ReSPPINN segmentations closely align with the DWI reference segmentation. Appendix B presents more qualitative results. Table 1 lists the average Dice score and volumetric agreement with acute DWI reference segmentations for ReSPPINN and the two commercial software packages. Dice scores of ReSPPINN demonstrate a significant improvement compared to Syngo.via and StrokeViewer. Unlike StrokeViewer and Syngo.via, which missed several smaller infarcts, our method successfully detected each one. Furthermore, there is only a marginal decline in the Dice score ($-0.02$) when using the ReSPPINN@500 CBF map compared to ReSPPINN-no-init@5000. ReSPPINN@500 outperforms Syngo.via and StrokeViewer, but also ReSPPINN-no-init@5000, in terms of volumetric agreement, with a mean difference closer to zero and a reduced absolute difference. Appendix C presents an analysis of Bland-Altman figures supporting these results.

Table 1: Infarct core segmentation results. Dice and mean or absolute volumetric difference (MVD, AVD), and false negative rate (FNR). We report mean (standard deviation) for 5 seeds. Symbols indicate if larger ($\uparrow$), smaller ($\downarrow$), or close to zero (0) values denote better performance.

| Method | Dice ($\uparrow$) | MVD, ml (0) | AVD, ml ($\downarrow$) | FNR ($\downarrow$) |
|---|---|---|---|---|
| SYNGO.VIA | 0.27 | $-24.1$ | 28.6 | 0.14 |
| STROKEVIEWER | 0.26 | $-23.3$ | 26.3 | 0.43 |
| ReSPPINN-no-init@5000 | 0.51(0.01) | 9.2(2.6) | 14.4(1.7) | 0.00(0.00) |
| ReSPPINN@500 | 0.49(0.02) | $-0.2$(2.2) | 14.0(1.1) | 0.00(0.00) |

## 5. Discussion and Conclusion

We presented ReSPPINN, a method for fast volume-based CT perfusion analysis using physics-informed neural fields. Our experiments show that hash-encodings help the neural fields to rapidly fit high-frequency details in tissue attenuation data and produce accurate perfusion maps. SPPINN operates on 2D axial slices rather than the entire volume and takes more than 40 minutes for full-volume perfusion map generation. ReSPPINN-no-init, on the other hand, achieves accurate perfusion maps in 12 minutes. Using the proposed meta-learned initialization, the networks converge faster and with greater stability, allowing for full-volume perfusion map generation within 1.2 minutes, which makes ReSPPINN suitable to be used in clinical practice. The strong bias introduced by the meta-learned initialization could be a disadvantage since ReSPPINN will be more likely to converge to a local rather than a global minimum. The segmentation performance and the strong visual agreement between the CBF from ReSPPINN@500 and ReSPPINN-no-init@5000 suggests, however, that acceleration does not harm the perfusion map quality and infarct detection accuracy, and only marginally affects the segmentation results in practice. Lastly, we show that the proposed method shows improved infarct core segmentation performance compared to commercial software. Using standard U-Net, our performance is on par with the top methods in the Ischemic Stroke Lesion Segmentation 2018 challenge, where the leading model achieved a Dice of 0.51 (Hakim et al., 2021) (see also Appendix E). We use U-Net since we aimed to show that ReSPPINN perfusion maps are effective for infarct core segmentation. Future work could investigate whether other approaches could further enhance the infarct core segmentation performance.

The main limitation of this study is the limited acute DWI data that was available for training and evaluation. Acute DWI imaging is not often acquired and therefore such reference segmentations are hard to come by. Another limitation is that the initialization works most efficiently if the data are pre-registered to a standard coordinate system.

In conclusion, we showed that meta-learning allows ReSPPINN to achieve rapid full-volume perfusion map generation in 1.2 minutes without compromising map quality. This computation time is brief enough to potentially enable future clinical use. Moreover, ReSPPINN achieves accurate infarct core segmentation outperforming commercial software.

## Acknowledgments

This work was part of the Artificial Intelligence for Early Imaging-Based Patient Selection in Acute Ischemic Stroke (AIRBORNE) project. This project was supported by Top Sector Life Sciences & Health and Nicolab B.V. The CLEOPATRA healthcare evaluation study was funded by Leading the Change (LtC). LtC is financed by Zorgverzekeraars Nederland (ZN) and supports various healthcare evaluations in the Netherlands as part of the Healthcare Evaluation Netherlands project and the CONTRAST consortium. The CONTRAST consortium acknowledges the support from the Netherlands Cardiovascular Research Initiative, an initiative of the Dutch Heart Foundation (CVON2015-01: CONTRAST), and from the Brain Foundation Netherlands (HA2015.01.06). The collaboration project is additionally financed by the Ministry of Economic Affairs by means of the PPP Allowance made available by the Top Sector Life Sciences & Health to stimulate public-private partnerships (LSHM17016). The CONTRAST consortium was funded in part through unrestricted funding by Stryker, Medtronic and Cerenovus. The funding sources were not involved in study design, monitoring, data collection, statistical analyses, interpretation of results, or manuscript writing.

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

# Appendix A. Implementation details

**SPPINN and ReSPPINN** Table 2 lists implementation details for SPPINN and our proposed ReSPPINN.

**Loss functions** Table 3 lists a description of the loss functions $\mathcal{L}_{TAC}$, $\mathcal{L}_{AIF}$, and $\mathcal{L}_{ODE}$. $\mathcal{L}_{TAC}$, $\mathcal{L}_{AIF}$, and $\mathcal{L}_{ODE}$ are equally weighted in the total loss function. In preliminary experiments, we empirically set the batch size to 25000. Adjusting the batch size primarily influences compute time, with minimal visual impact on the quality of the perfusion maps.

**Hash-encoding** Figure 4 presents an illustrative example of the hash-encoding operation. In the example, we demonstrate hash-encoding with two resolutions ($L = 2$) in a two-dimensional setting, while our method actually utilizes $L = 16$ resolutions within the three-dimensional domain. The multi-resolution hash-encoding operation divides the domain into multiple grids, with each grid point indexed by an integer. For each grid point, we assign $d = 2$ learnable weights in a hash table per resolution, which can be retrieved by looking up the integer index. For a spatial coordinate $\mathbf{x}$, the hash-encoding operation identifies the weights for the nearest grid points in the hash table for each resolution. Per resolution, the weights corresponding to the grid points are then linearly interpolated according to the relative positions of the coordinate with respect to the grid points. The interpolated weights for each resolution are subsequently stacked to obtain an embedding $e \in \mathbb{R}^{Ld}$. This embedding is then used as the input to the networks. We used a PyTorch implementation of multi-resolution hash-encodings (Hsiao, 2023).

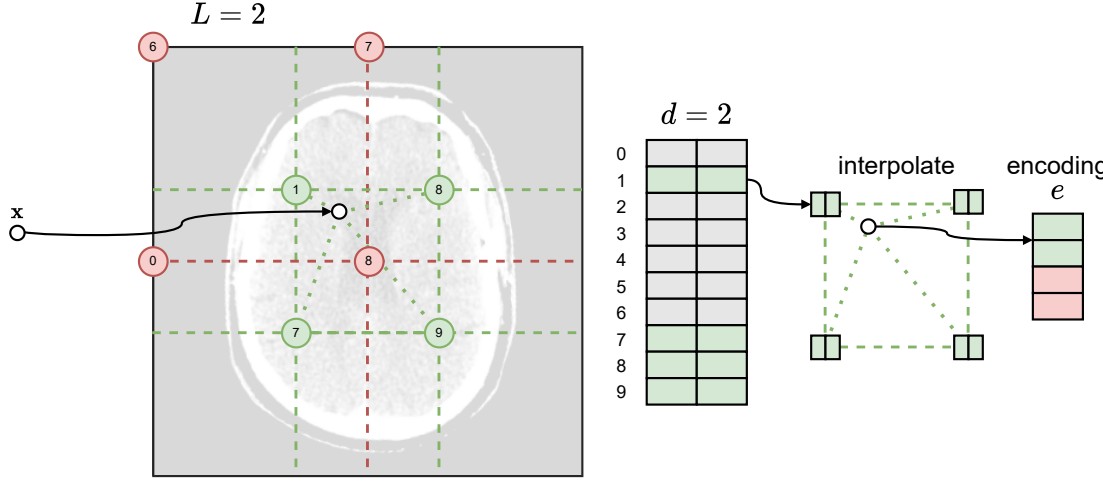

Figure 4: Illustrative example of the hash-encoding operation. Figure inspired by Müller et al. (2022).

**Reptile meta-learning** Table 4 lists the implementation details for Reptile meta-learning. Figure 5 presents the meta-learned initializations for $f_{\text{TAC}}(\mathbf{x}, t; \phi)$ at two time points and for $f_{\text{ODE}}(\mathbf{x}; \xi)$ for the CBF perfusion parameter. We note that the initializations already resemble brain-like attenuation or blood flow values. For instance, there is a noticeable increased attenuation in the later timepoint for $f_{\text{TAC}}(\mathbf{x}, t; \phi)$'s initialization. Similarly, the initial CBF pattern shows characteristic features, such as increased flow near supplying arteries. Figure 6 presents the output of $f_{\text{ODE}}(\mathbf{x}; \xi)$ for the CBF at various iterations for ReSPPINN@500

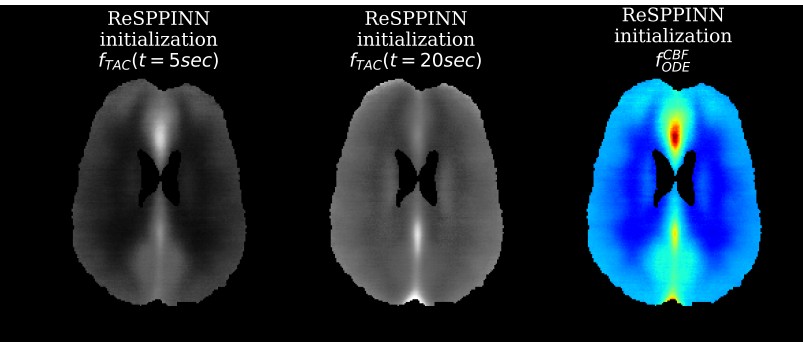

Figure 5: Meta-learned initializations for $f_{\text{TAC}}(\mathbf{x}, t; \phi)$ at two time points and for $f_{\text{ODE}}(\mathbf{x}; \xi)$ (CBF is shown).

starting from the meta-learned initialization. Moreover, it shows the CBF for ReSPPINN-no-init@5000 for visual comparison. It illustrates $f_{\text{ODE}}(\mathbf{x}; \xi)$'s iterative progress in achieving a closer fit to the data and accurately determining the perfusion parameters.

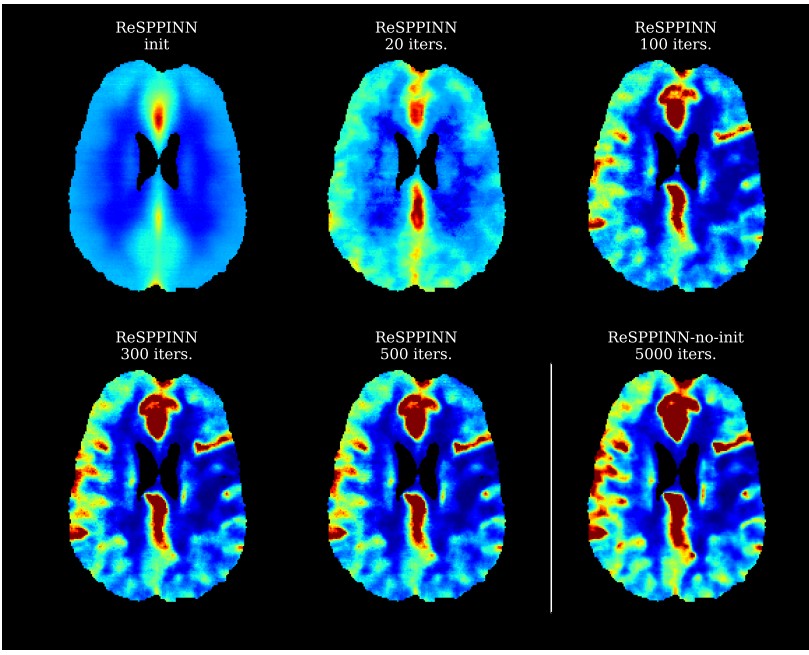

Figure 6: Starting from the meta-learned initialization ReSPPINN@500 iteratively learns the perfusion map. ReSPPINN-no-init@5000 shown for comparison.

**Segmentation model**   Table 5 lists the implementation details for training the U-Net segmentation model. We use the U-Net architecture within the MONAI framework (Cardoso et al., 2022). Furthermore, we employ augmentations and loss functions from MONAI. Because we select the model based on only one validation scan, we keep a running mean over the last ten epochs and use the mean Dice score to select our best model.

## Appendix B. Additional qualitative results

Figure 7 presents qualitative results for five patients. It includes CBF maps for SPPINN@5000 and ReSPPINN@500, along with DWI images, core segmentations from ReSPPINN, and reference segmentations. Similar to Figure 3, there are significant differences between the SPPINN and ReSPPINN CBF maps. The infarct is visible on SPPINN's smoother perfusion maps, but distinguishing brain structures is more challenging. The infarct is also visible on ReSPPINN's CBF map and the qualitative structural similarity with the DWI is larger than for SPPINN. The predicted infarct core segmentation and reference show generally good overlap. The results for two patients with small infarcts are displayed in the second and last row of Figure 7. Our approach missed not a single infarct, unlike STROKEVIEWER and SYNGO.VIA, which failed to identify several infarcts. This is also reflected by the false negative rate in Table 1. Specifically, SYNGO.VIA missed 2 out of 14 infarcts, and STROKEVIEWER missed 6 out of 14, often failing to identify particularly small infarcts. This underscores our model's strength in detecting smaller infarcts, which are often overlooked by alternative approaches.

## Appendix C. Bland–Altman

Figure 8 shows Bland-Altman figures for the two commercially available CTP software packages and ReSPPINN with and without Reptile acceleration. For both commercial methods, the average difference between the predicted and the reference standard infarct volume is larger than for ReSPPINN. For the commercial methods, we observe a negative bias, suggesting an underestimation of the predicted infarct core volume. The negative bias seems primarily due to measurements over 50 ml since we observe a negative trend for larger infarct volumes. For ReSPPINN@500, we observe little to no bias, also for larger infarct volumes. ReSPPINN-no-init@5000, however, shows a larger bias, probably caused by an outlier with a considerable overestimation of the infarct core volume. For all methods, we observe increased variability in the difference as the mean infarct volume grows. The limits of agreement are narrowest for ReSPPINN@500. In conclusion, ReSPPINN@500 has the best volumetric correspondence to reference standard DWI infarct core volume estimations.

## Appendix D. Segmentation results with slice-based SPPINN perfusion maps

Table 6 lists the infarct core segmentation results using the perfusion maps generated by the original slice-based SPPINN. This experiment resulted in a Dice score of 0.50, an MVD of 10.2 ml, an AVD of 17.3 ml, and an FNR of 0.00. Though the performance is similar to the proposed ReSPPINN approach in terms of Dice score and FNR, the volumetric agreement with the reference DWI segmentation is significantly worse, suggesting an overestimation

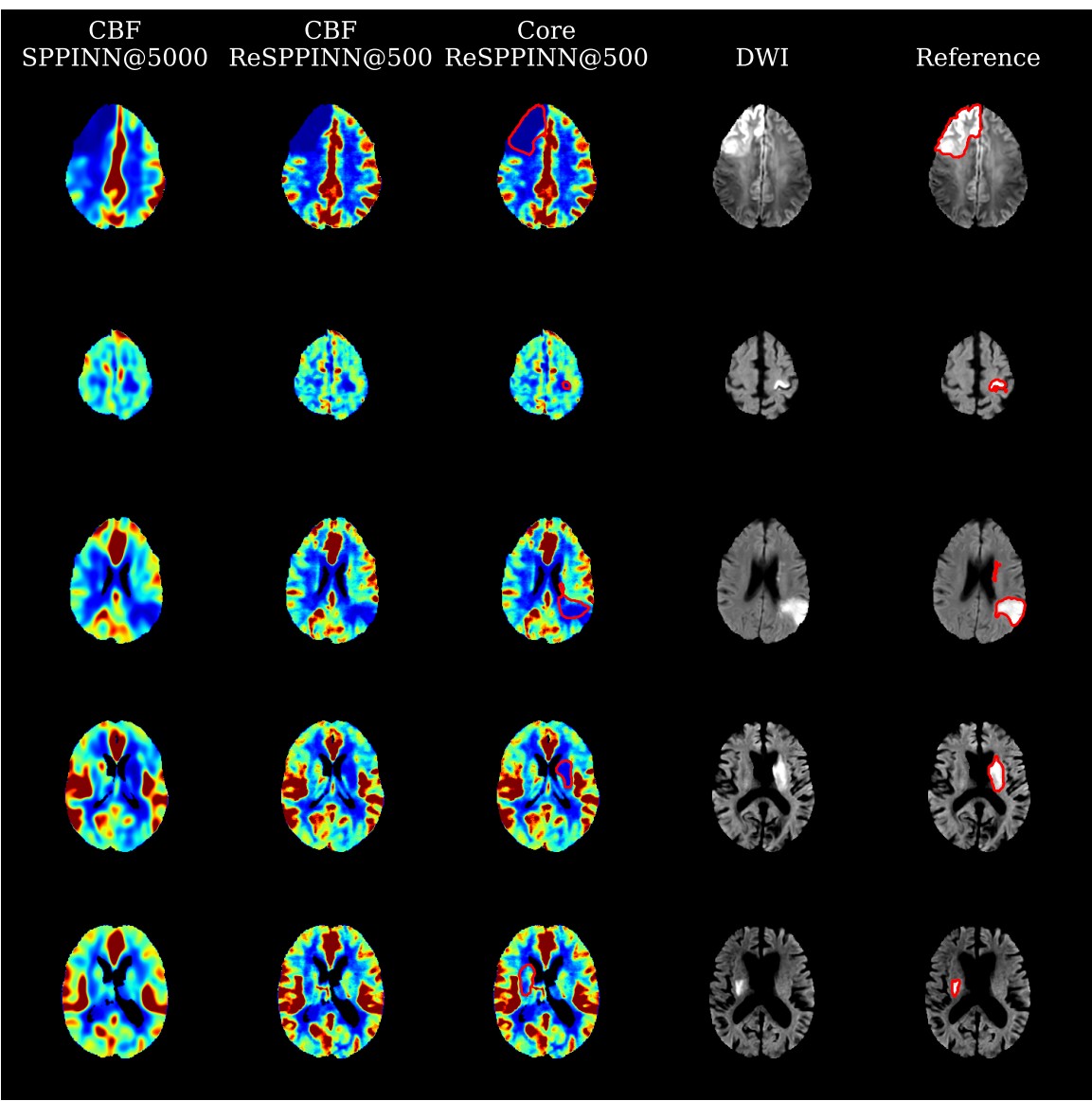

Figure 7: Comparison of CBF maps, DWI, and core segmentations for five patients, showcasing the differences between SPPINN@5000 and ReSPPINN@500.

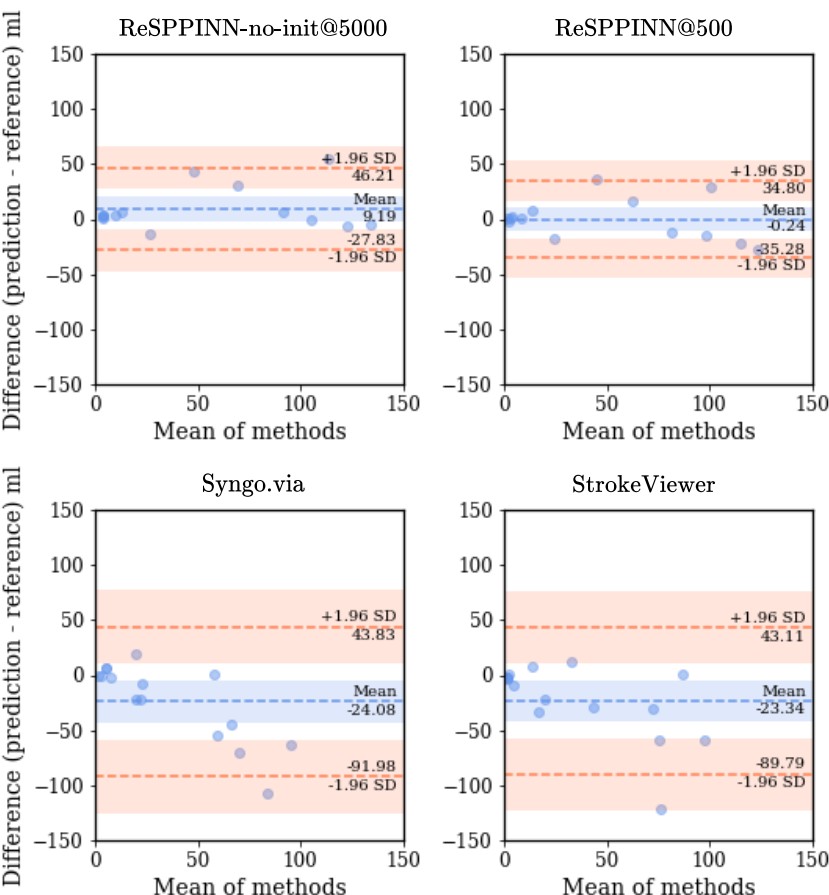

Figure 8: Bland-Altman figures for ReSPPINN with and without initialization and the two commercial software packages.

Table 2: Implementation configuration for SPPINN and ReSPPINN.

| | SPPINN | ReSPPINN |
|---|---|---|
| config | value | value |
| spatial input dimensions | 2D | 3D |
| number of layers | | |
| $\quad f_{\text{TAC}}(\mathbf{x}, t; \phi)$ | 3 | 3 |
| $\quad f_{\text{AIF}}(t; \theta)$ | 3 | 3 |
| $\quad f_{\text{ODE}}(\mathbf{x}; \xi)$ | 3 | 3 |
| neurons per layer | | |
| $\quad f_{\text{TAC}}(\mathbf{x}, t; \phi)$ | 128 | 16 |
| $\quad f_{\text{AIF}}(t; \theta)$ | 16 | 16 |
| $\quad f_{\text{ODE}}(\mathbf{x}; \xi)$ | 64 | 16 |
| activation function | | |
| $\quad f_{\text{TAC}}(\mathbf{x}, t; \phi)$ | Siren $w = 15, w_0 = 15$ | Siren $w = 15, w_0 = 15$ |
| $\quad f_{\text{AIF}}(t; \theta)$ | Siren $w = 1, w_0 = 1$ | Siren $w = 1, w_0 = 1$ |
| $\quad f_{\text{ODE}}(\mathbf{x}; \xi)$ | Siren $w = 15, w_0 = 15$ | Siren $w = 15, w_0 = 15$ |
| $\quad f_{\text{ODE}}(\mathbf{x}; \xi)$, last layer | Exp | Exp |
| optimizer | Adam | Adam |
| base learning rate | 1e-3 | 1e-3 |
| learning rate schedule | OneCycleLR | OneCycleLR |
| warm-up iterations | 0 | 0 |
| batch size ($B$) | 25000 | 25000 |
| hash-encoding | ✗ | ✓ |
| $\quad$ levels | - | 16 |
| $\quad$ features per level | - | 2 |
| $\quad \log_2$ hashmap size | - | 15 |
| $\quad$ base resolution | - | 16 |
| $\quad$ finest resolution | - | 4096 |
| GPU memory requirement (MB) | 1134 | 1670 |

Table 3: Description and implementation of the loss function.

| loss | function | description |
|------|----------|-------------|
| $\mathcal{L}_{TAC}$ | $\frac{1}{|B|}\sum_{(\mathbf{x},t)}\big\|f_{\text{TAC}}(\mathbf{x},t) - C_{TAC}(\mathbf{x},t)\big\|$ | The goal for $f_{\text{TAC}}(\mathbf{x},t;\phi)$ is to fit the observed tissue attenuation $C_{TAC}(\mathbf{x},t)$ at all voxel locations. $\mathcal{L}_{TAC}$, therefore, is the $l_1$-norm of the difference between the predicted and observed tissue attenuation. The observed data are discretely sampled at spatio-temporal coordinates $(\mathbf{x},t)$. At each iteration, we sample a subset $B$ coordinates from all spatio-temporal coordinates to compute $\mathcal{L}_{TAC}$. |
| $\mathcal{L}_{AIF}$ | $\frac{1}{|B|}\sum_{t}\big\|f_{\text{AIF}}(t) - C_{\text{AIF}}(t)\big\|$ | The goal for $f_{\text{AIF}}(t;\theta)$ is to fit the observed arterial input function $C_{\text{AIF}}(t)$. $\mathcal{L}_{AIF}$, therefore, is the $l_1$-norm of the difference between the predicted and observed arterial input function. The observed data are discretely sampled at temporal coordinates $t$. At each iteration, we sample a subset $B$ spatio-temporal coordinates from all spatio-temporal coordinates and use only the sampled temporal coordinates to compute $\mathcal{L}_{AIF}$. |
| $\mathcal{L}_{ODE}$ | $\frac{1}{|B|}\sum_{(\mathbf{x},\tau)}\big\|r(\mathbf{x}_v,\tau)\big\|$ | The goal for $f_{\text{ODE}}(\mathbf{x};\xi)$ is to obtain the best estimate of the perfusion parameters at all voxel locations. $\mathcal{L}_{ODE}$, therefore, is the $l_1$-norm of the residual equation Equation (4). The observed tissue intensity data are discretely sampled at spatio-temporal coordinates $(\mathbf{x},t)$. For $\mathcal{L}_{ODE}$, however, we *continuously* sample temporal coordinates $(\mathbf{x},\tau)$ within the temporal domain. At each iteration, we use the subset $B$ of coordinates $(\mathbf{x},\tau)$ to compute $\mathcal{L}_{ODE}$. |

Table 4: Implementation configuration for Reptile meta-learning.

| config | value |
|--------|-------|
| outer loop | |
| optimizer | Adam |
| base learning rate | 1e-2 |
| learning rate schedule | OneCycleLR |
| warm-up iterations | 30% |
| $N_{out}$ iterations | 7500 |
| $\epsilon$ | 0.1 |
| inner loop | |
| optimizer | Adam |
| base learning rate | 1e-3 |
| learning rate schedule | OneCycleLR |
| warm-up iterations | 0% |
| $N_{in}$ iterations | 500 |

Table 5: Implementation configuration for the proposed U-Net.

| config | value |
|---|---|
| features per layer | 16, 16, 32, 32, 64 |
| augmentations | RandFlip, RandRotate, RandZoom |
| optimizer | Adam |
| base learning rate | 5e-5 |
| learning rate schedule | OneCycleLR |
| warm-up iterations | 0 |
| loss function | DiceCELoss(lambda_dice=1, lambda_ce=1) |
| batch size | 1 |
| epochs | 500 |
| patch size | 256x256x20 |
| patch stichting | SlidingWIndowInferer(mode=Gaussian, overlap=0.5) |
| post-processing | Infarct core restricted to affected hemisphere |

Table 6: Infarct core segmentation results for segmentation model using SPPINN and ReSPPINN perfusion maps. Dice and mean or absolute volumetric difference (MVD, AVD), and false negative rate (FNR). We report mean (standard deviation) for 3 seeds. Symbols indicate if larger (↑), smaller (↓), or close to zero (0) values denote better performance.

| Method | Dice (↑) | MVD, ml (0) | AVD, ml (↓) | FNR (↓) |
|---|---|---|---|---|
| SPPINN@5000 (slice-based) | 0.50(0.01) | 10.2(1.4) | 17.3(0.7) | 0.00(0.00) |
| ReSPPINN@500 (volume-based) | 0.49(0.02) | −0.2(2.2) | 14.0(1.1) | 0.00(0.00) |

of the infarct core. This may be a result of a higher level of smoothness in the slice-based perfusion maps, leading to poorly defined segmentation boundaries. Alternatively, the absence of 3D context, and therefore, structural differences between slices, may affect results.

It should further be noted that the high difference in computational time (43 minutes for SPPINN, 1.2 minutes for ReSPPINN) makes SPPINN unusable in clinical practice. The slice-wise approach further ignores 3D context, which is supported by our proposed method. For a slice-based approach, fitting perfusion values in the brain's top and bottom parts is challenging and prone to suboptimal optimization, due to the smaller brain area in those regions and the fact that those regions are more prone to image artifacts. A full-volume approach mitigates this by calculating loss across the entire volume, enhancing stability and guiding optimization. The volume-based approach therefore offers significant advantages.

## Appendix E. Ischemic Stroke Lesion Segmentation Challenge 2018 challenge results

Table 7 lists the top-five results of the Ischemic Stroke Lesion Segmentation Challenge (ISLES) 2018 (Hakim et al., 2021), as detailed in the official leaderboard at https://www.smir.ch/ISLES/Start2018. The goal of the challenge was to segment the acute phase DWI MRI reference infarct core from CT and CTP imaging. The available data (63 patients for training, 40 patients for testing) comprised baseline non-contrast CT, CTP source data, and four perfusion maps (CBF, CBV, MTT, Tmax) generated by RAPID (RAPID; iSchemaview, Menlo Park, CA). The reference standard was segmented on acute phase DWI MRI with a median time between CT and MRI of 36 minutes. Most participants of the challenge employed the non-contrast CT and four perfusion maps (CBF, CBV, MTT, Tmax) as inputs to the segmentation models. We refer to Hakim et al. (2021) for a full overview of the challenge and main results. The Dice score of ReSPPINN is competitive, aligning with the top-five methods featured on the ISLES 2018 leaderboard. Notably, for ReSPPINN, the absolute volumetric difference is approximately $3 - 5$ ml higher.

Table 7: Top-five results of the Ischemic Stroke Lesion Segmentation Challenge (ISLES) 2018 (Hakim et al., 2021).

| Team | Reference | Dice ($\uparrow$) | AVD, ml ($\downarrow$) |
|------|-----------|------|---------|
| songt1 | Song and Huang (2019) | 0.51 | 10.24 |
| clera2 | Clèrigues et al. (2019) | 0.49 | 12.18 |
| ghosp1 | N/A | 0.49 | 9.30 |
| zhans10 | N/A | 0.49 | 9.81 |
| pengl1 | Liu (2019) | 0.49 | 10.08 |

