# OpenReview forum: "Accelerating physics-informed neural fields for fast CT perfusion analysis in acute ischemic stroke"
_MIDL.io/2024/Conference — MIDL 2024 Oral_

### Official Review · Reviewer_pBkj · 2024-02-25

**Confidence:** 5
**Preliminary Rating:** 5
**Recommendation:** Best Paper Award
**Final Rating:** 5

**Summary:**

The authors of this work present a substantial improvement over a previously proposed algorithm for CT perfusion analysis in acute ischemic stroke and that is based on spatiotemporal physics-informed neural fields (SPPINN). Specifically, the authors extend the preceding 2D method and enable fast and accurate 3D estimations of cerebral blood flow (CBF) maps. Key to reduce the training / optimization time from >40 mins to 1.2mins and to produce visually higher-fidelity 3D maps compared to the 2D SPPINN are multi-resolution hash-encodings and Reptile meta-learning. Finally, to demonstrate the quality of the reconstructed CBF maps, a segmentation model was trained and evaluated on the output of the proposed method and commerical software in a leave-one-out-crossvalidation over 15 patients.

**Strengths:**

- The paper is very well written and interesting to read
- The method is novel in that it builds on top of important advances in the field
- The choice of algorithms is very well motivated and explained
- Despite the limited amount of ground truth data, the evaluation seems appropriate and sound. The results are convincing

**Weaknesses:**

- Even though the main contribution is ReSPINN for estimating CBF maps, the quantitative performance evaluation mostly focuses on the segmentation task. In particular, since the original dataset holds 898 patients with CTP from STROKEVIEWER, it may be insightful to understand how similar/dissimilar the maps are. In that regard, it may be interesting to also compute and compare the residual loss for the STROKEVIEWER, Syngo.Via CBF, ReSPINN maps.

- The segmentation model was not trained/evaluated on the original SPPINN output

- Qualitative results are only shown for a single patient

**Detailed Comments:**

- Neural field training: Could the authors please comment on how a change in the number of collocation points may impact the quality of the results?
- Figure 2: Could the authors please explain why the loss curves of SPPINN@5000 and @10000 are different? Was a different random seed used?

**Justification Of Final Rating:**

I would like to thank the authors for addressing all of my questions and concerns in such detail. In addition, the authors have made significant improvements to the already very good manuscript and the new insights further strengthen the findings. I am absolutely certain that this submission will be of great interest to the MIDL community.

**Justification Of The Preliminary Rating:**

From my point of view, the presented paper has only very few minor weaknesses, which could be easily integrated in the rebuttal and without changing the core of the paper. I believe that this manuscript is of particular value to the community, in particular because the proposed method is novel, significantly faster, and produces higher fidelity CBF maps.

**Questions To Address In The Rebuttal:**

In addition to the points raised under weaknesses, could the authors please comment on the memory requirements of SPPINN and ReSPPINN? Would it be possible to apply SPPINN in parallel? What speed-up would I expect?

**Special Issue:**

Yes

---

> ### Author Response · Authors · 2024-03-17
>
> We thank the reviewer for their review, helpful comments, and suggestions.
>
> **W1:** Even though the main contribution is ReSPINN for estimating CBF maps, the quantitative performance evaluation mostly focuses on the segmentation task. In particular, since the original dataset holds 898 patients with CTP from STROKEVIEWER, it may be insightful to understand how similar/dissimilar the maps are. In that regard, it may be interesting to also compute and compare the residual loss for the STROKEVIEWER, Syngo.Via CBF, ReSPINN maps.
>
> **ANSWER:**
>
> We understand that the missing evaluation on the computed perfusion values limits the evaluation of our method. Unfortunately, there exist no ground truth perfusion values, and therefore, it is not possible to compute the accuracy of the values themselves. Furthermore, it is also not possible to calculate a meaningful residual loss for the perfusion maps generated by other software. For one, it's important to mention that the optimization objectives of these software packages, often employing non-parametric approaches, might have differed from our parametric approach. As a result, these baseline software packages did not use an ODE for inferring perfusion parameters, making the calculation of the residual loss for these applications not applicable. Secondly, the perfusion maps from StrokeViewer, to take as an example, consist of relative perfusion maps where all values are compared to the healthy hemisphere. However, the exact method of how this relativity is computed is unknown, and we cannot reconstruct the absolute perfusion values. Thus, we cannot input these relative values into the loss equation. Thirdly, to calculate the residual loss, we average over collocation points, but we have no means to calculate \\(dC_{tac}(t)/dt\\) for these other software packages. Therefore, calculating the residual loss is not possible. In our previous work\*, we employed simulated data to assess the accuracy of the perfusion estimates and showed improved accuracy in comparison to other (commercial) methods. The focus of this follow-up study was on making the algorithm relevant for practical clinical practice, and hence, on speeding up perfusion map generation and enhancing infarct estimations, which is why we have limited our evaluation to these aspects. To conclude, since no ground truth can exist, we must unfortunately rely on a qualitative evaluation, and quantitatively, on evaluating the performance on downstream tasks.
>
> \*_Spatio-temporal physics-informed learning: A novel approach to CT perfusion analysis in acute ischemic stroke_, Medical Image Analysis, Volume 90, 2023)
>
> **CHANGES IN MANUSCRIPT:**
>
> In the baseline description in Section 2.5, we add:
>
> “For a review of the vanilla _SPPINN_ implementation _and its quantitative performance_, we refer to De Vries et al. (2023b).”
>
> In the Appendix B Figure 7, we add more qualitative results for ReSPPINN including a discussion:
>
> _Appendix B. Additional qualitative results_
>
> “_Figure 7 presents qualitative results for five patients. It includes CBF maps for SPPINN@5000 and ReSPPINN@500, along with DWI images, core segmentations from ReSPPINN, and reference segmentations. Similar to Figure 3, there are significant differences between the SPPINN and ReSPPINN CBF maps. The infarct is visible on SPPINN’s smoother perfusion maps, but distinguishing brain structures is more challenging. The infarct is also visible on ReSPPINN's CBF map and the qualitative structural similarity with the DWI is larger than for SPPINN. The predicted infarct core segmentation and reference show generally good overlap. The results for two patients with small infarcts are displayed in the second and last row of Figure 7. Our approach missed not a single infarct, unlike StrokeViewer and Syngo.via, which failed to identify several infarcts. This is also reflected by the false negative rate in Table 1. Specifically, Syngo.via missed 2 out of 14 infarcts, and StrokeViewer missed 6 out of 14, often failing to identify particularly small infarcts. This underscores our model's strength in detecting smaller infarcts, which are often overlooked by alternative approaches._”

---

> ### Author Response · Authors · 2024-03-17
>
> **W2: The segmentation model was not trained/evaluated on the original SPPINN output**
>
> **ANSWER:** We trained and evaluated the segmentation model also on the SPPINN perfusion maps. We ran the experiment for three seeds, resulting in a Dice score of 0.50 and a MVD and AVD of 10.2 ml and 17.3 ml, respectively. The FNR was 0.00. In terms of Dice score and FNR the segmentation model on SPPINN perfusion maps performs similarly as the proposed ReSPPINN. However, the volumetric agreement with reference DWI segmentations is poorer than for ReSPPINN, suggesting overestimation of the infarct core. We hypothesize this may stem from the smoother perfusion map, leading to less defined segmentation boundaries for the model. Alternatively, the absence of 3D context and, therefore, structural differences between slices, could have affected results.
>
> In addition to the worse volumetric agreement, it is important to highlight the computational time difference for perfusion map generation: the slice-based approach took approximately 43 minutes while volume-based ReSPPINN took only 1.2 minutes. Moreover, the slice-based method neglects spatial information from surrounding slices. The aim of this paper was to leverage the volume-based data as a whole. Moreover, fitting perfusion values in the brain's top and bottom parts is challenging and prone to suboptimal optimization, due to the smaller brain area in those regions and the fact that those regions are more prone to image artifacts. A full-volume approach mitigates this by calculating loss across the entire volume, enhancing stability and guiding optimization. Therefore, the volume-based approach offers significant advantages.
>
> **CHANGES IN MANUSCRIPT:**
>
> We add in Appendix D Table 6 with infarct core segmentation results for slice-based SPPINN perfusion maps, accompanied by the following text:
>
> “_Table 6 lists the infarct core segmentation results using the perfusion maps generated by the original slice-based SPPINN. This experiment resulted in a Dice score of 0.50, an MVD of 10.2 ml, an AVD of 17.3 ml, and an FNR of 0.00. Though the performance is similar to the proposed ReSPPINN approach in terms of Dice score and FNR, the volumetric agreement with the reference DWI segmentation is significantly worse, suggesting an overestimation of the infarct core. This may be a result of a higher level of smoothness in the slice-based perfusion maps, leading to poorly defined segmentation boundaries. Alternatively, the absence of 3D context, and therefore, structural differences between slices, may affect results. It should further be noted that the high difference in computational time (43 minutes for SPPINN, 1.2 minutes for ReSPPINN) makes SPPINN unusable in clinical practice. The slice-wise approach further ignores 3D context, which is supported by our proposed method. For a slice-based approach, fitting perfusion values in the brain's top and bottom parts is challenging and prone to suboptimal optimization, due to the smaller brain area in those regions and the fact that those regions are more prone to image artifacts. A full-volume approach mitigates this by calculating loss across the entire volume, enhancing stability and guiding optimization. The volume-based approach therefore offers significant advantages._”

---

> ### Author Response · Authors · 2024-03-17
>
> **W3:** Qualitative results are only shown for a single patient
>
> **ANSWER and CHANGES IN MANUSCRIPT:** We include additional ReSPPINN qualitative results in the Appendix B Figure 7 for several patients, including a discussion:
>
> _Appendix B. Additional qualitative results_
>
> “_Figure 7 presents qualitative results for five patients. It includes CBF maps for SPPINN@5000 and ReSPPINN@500, along with DWI images, core segmentations from ReSPPINN, and reference segmentations. Similar to Figure 3, there are significant differences between the SPPINN and ReSPPINN CBF maps. The infarct is visible on SPPINN’s smoother perfusion maps, but distinguishing brain structures is more challenging. The infarct is also visible on ReSPPINN's CBF map and the qualitative structural similarity with the DWI is larger than for SPPINN. The predicted infarct core segmentation and reference show generally good overlap. The results for two patients with small infarcts are displayed in the second and last row of Figure 7. Our approach missed not a single infarct, unlike StrokeViewer and Syngo.via, which failed to identify several infarcts. This is also reflected by the false negative rate in Table 1. Specifically, Syngo.via missed 2 out of 14 infarcts, and StrokeViewer missed 6 out of 14, often failing to identify particularly small infarcts. This underscores our model's strength in detecting smaller infarcts, which are often overlooked by alternative approaches._”
>
> **DC1:** Neural field training: Could the authors please comment on how a change in the number of collocation points may impact the quality of the results?
>
> **ANSWER:** We empirically set the batch size to 25000 in preliminary experiments. Changing the batch size (and with that the number of sampled collocation points per iteration) primarily changes the compute time if we keep the maximum number of iterations fixed. For ReSPPINN@500, the compute times for batch sizes of 5000, 25000, and 75000, are 0.9, 1.2, and 2.6 minutes, respectively. We do not observe visual differences in the perfusion maps.
>
> **CHANGES IN MANUSCRIPT:**
>
> In Appendix A, Loss functions, we add:
>
> “_In preliminary experiments, we empirically set the batch size to 25000. Adjusting the batch size primarily influences compute time, with minimal visual impact on the quality of the perfusion maps._”
>
> **DC2:** Figure 2: Could the authors please explain why the loss curves of SPPINN@5000 and @10000 are different? Was a different random seed used?
>
> **ANSWER:** The learning rate scheduler (OneCycleLR) calculates the scheduled change to the learning rate as a function of the maximum number of iterations. This causes some variation in the optimization process between SPPINN@5000 and SPPINN@10000.
>
> **CHANGES IN MANUSCRIPT:**
>
> in Results, Effectiveness of hash-encodings, we add:
>
> “_Optimization differences between SPPINN@5000 and @10000 stem from our iteration-based learning rate scheduler._”
>
> **Q1:** In addition to the points raised under weaknesses, could the authors please comment on the memory requirements of SPPINN and ReSPPINN? Would it be possible to apply SPPINN in parallel? What speed-up would I expect?
>
> **ANSWER:** The GPU memory requirement for volume-based ReSPPINN is 1670MB. For slice-based SPPINN, the memory requirement is 1134MB. Parallelization is definitely an option. In our experiments with process parallelization, however, we found that the strongest limit on acceleration is imposed by the GPU utilization. Unfortunately, we were not able to implement a multi-GPU set-up. The total compute time for a full volume with slice-based SPPINN was about 10 minutes on our NVIDIA V100 GPU.
>
> The limitation of this parallelized slice-wise approach, however, is that spatial information from surrounding slices is neglected. Fitting perfusion values in the brain's top and bottom parts is challenging due to the smaller brain area in those regions and the fact that those regions are more prone to image artifacts, which may compromise optimization for slice-based approaches. A full-volume approach mitigates this by calculating loss across the entire volume, enhancing stability and guiding optimization. Additionally, representing the entire volume as a neural field supports downstream tasks like segmentation through neural field weights (e.g. with the recently proposed inr2vec\*), and enables benefits like super-resolution, unattainable with a slice-wise method. Therefore, the volume-based approach offers significant advantages.
>
> \*) De Luigi, Luca et al. 2023. "Deep Learning on Implicit Neural Representations of Shapes." In Proceedings of the International Conference on Learning Representations (ICLR).
>
> **CHANGES IN MANUSCRIPT:**
>
> We add the memory requirements in Appendix A, Table 2.

---

### Official Review · Reviewer_UsiZ · 2024-02-28

**Confidence:** 5
**Preliminary Rating:** 4
**Recommendation:** Poster

**Summary:**

The paper presents a neural field representation combined with a physics based modeling approach to extract the perfusion and blood flow in contrast CT perfusion scans. The presented method is novel and relevant for performing perfusion analysis. However, the presented accuracy evaluation is somewhat limited, particularly in terms of assessing quality of computed perfusion maps. The method involves several different components and a lot of these components are not well explained, making it difficult to understand how the method is implemented in practice.

**Strengths:**

* A mathematically elegant methodology that combines neural field representation with physics based modeling to model the dynamics of contrast perfusion and blood flow for stroke images.
* The mathematics behind the approach is well explained.
* Results are presented for stroke volume segmentation as well as in evaluating the computational efficiency of the approach

**Weaknesses:**

* Analysis is performed using leave one out cross validation for the stroke segmentation using a relatively small number of patients, raising concerns regarding generalization capabilities. Presenting where the approach works and where it doesn't work would be useful in this regard to understand the limits of the methodology.
* Accuracy of the perfusion imaging is limited to stroke segmentation but the image quality and the accuracy of the computed perfusion values themselves are not compared.
* Details of the meta learning approach as well as other aspects of the learning methodology are somewhat limited. It would be helpful to see a more detailed explanation of the method at least in the appendix.

**Detailed Comments:**

Please see comments regarding strengths and weaknesses.

**Justification Of The Preliminary Rating:**

The paper is novel and the approach seems to be reasonably accurate for the stroke segmentation. The neural field representation with physics based modeling is technically novel and combination with the meta learning approach shows promise in terms of reducing the computational needs of the approach.

**Questions To Address In The Rebuttal:**

* Analysis is performed using leave one out cross validation for the stroke segmentation using a relatively small number of patients, raising concerns regarding generalization capabilities. Presenting where the approach works and where it doesn't work would be useful in this regard to understand the limits of the methodology.
* How is the visual quality of the perfusion image impacted by the hash coding approach? How does the accuracy of the computed perfusion values change without and with the hash coding method?

**Special Issue:**

No

---

> ### Author Response · Authors · 2024-03-16
>
> We appreciate the reviewer's feedback and valuable suggestions.
>
> **W1 & Q1:** Analysis is performed using leave one out cross validation for the stroke segmentation using a relatively small number of patients, raising concerns regarding generalization capabilities. Presenting where the approach works and where it doesn't work would be useful in this regard to understand the limits of the methodology.
>
> **ANSWER:** We recognize that the small size of our dataset limits our evaluation. To address this, we ran the experiments 4 additional times and updated Table 1 to include mean results - and the standard deviation of these results - across five runs with different seeds. Moreover, we added additional qualitative perfusion maps and segmentation results for several patients in Appendix B, with a discussion about the findings.
>
> **CHANGES IN MANUSCRIPT:**
>
> In the results, we update Table 1 with the average results over the five runs. Similarly, we update the Bland Altman plots in Figure 8. We also update Table 1 to include the false negative rate: our approach missed not a single infarct (FNR=0.00), unlike StrokeViewer (FNR=0.43) and Syngo.via (FNR=0.14), which failed to identify several infarcts.
>
> In introduction, we add:
>
> “Initially, we assess the segmentation _and detection_ performance of _ReSPPINN_ against that of two commercial CTP software packages.”
>
> In Experiments, we add:
>
> “We measure the average Dice score, mean volumetric difference (MVD), and absolute volumetric difference (AVD), between the reference and automatic segmentations, _and the false negative rate (FNR) for infarct detection._”
>
> In Results, we add:
>
> “_Appendix B presents more qualitative results._”
>
> In Results, we also add:
>
> “_Unlike StrokeViewer and Syngo.via, which missed several smaller infarcts, our method successfully detected each one._”
>
> In Table 1 caption, we add:
>
> “Dice and mean or absolute volumetric difference (MVD, AVD)**,** _and false negative rate (FNR). We report mean (standard deviation) for 5 seeds.”_
>
> In Discussion and Conclusion, we add:
>
> “… that acceleration does not harm the perfusion map quality _and infarct detection accuracy,_ and only marginally affects the segmentation results in practice.”
>
> In the Appendix B Figure 7, we add more qualitative results including a discussion:
>
> _Appendix B. Additional qualitative results_
>
> “_Figure 7 presents qualitative results for five patients. It includes CBF maps for SPPINN@5000 and ReSPPINN@500, along with DWI images, core segmentations from ReSPPINN, and reference segmentations. Similar to Figure 3, there are significant differences between the SPPINN and ReSPPINN CBF maps. The infarct is visible on SPPINN’s smoother perfusion maps, but distinguishing brain structures is more challenging. The infarct is also visible on ReSPPINN's CBF map and the qualitative structural similarity with the DWI is larger than for SPPINN. The predicted infarct core segmentation and reference show generally good overlap. The results for two patients with small infarcts are displayed in the second and last row of Figure 7. Our approach missed not a single infarct, unlike StrokeViewer and Syngo.via, which failed to identify several infarcts. This is also reflected by the false negative rate in Table 1. Specifically, Syngo.via missed 2 out of 14 infarcts, and StrokeViewer missed 6 out of 14, often failing to identify particularly small infarcts. This underscores our model's strength in detecting smaller infarcts, which are often overlooked by alternative approaches._”
>
> **W2:** Accuracy of the perfusion imaging is limited to stroke segmentation but the image quality and the accuracy of the computed perfusion values themselves are not compared.
>
> **ANSWER:** We understand that the missing evaluation on the computed perfusion values limits the evaluation of our method. The problem is, however, that there exists no such thing as ground truth perfusion values - there is no _gold standard_. Therefore, it is not possible to compute the accuracy of the values itself. In our previous work\*, we employed simulated data to assess the accuracy of the perfusion estimates and showed improved accuracy in comparison to other (commercial) methods. The focus of this follow-up study was on making the algorithm relevant for practical clinical practice, and hence, on speeding up perfusion map generation and enhancing infarct estimations. Our evaluation therefore prioritizes these aspects, since we believe the core estimations hold significant clinical value, with perfusion maps serving as means towards this end.
>
> \*_Spatio-temporal physics-informed learning: A novel approach to CT perfusion analysis in acute ischemic stroke_, Medical Image Analysis, Volume 90, 2023
>
> **CHANGES IN MANUSCRIPT:**
>
> In the baseline description in Section 2.5, we add:
>
> “For a review of the vanilla SPPINN implementation _and its quantitative performance_, we refer to De Vries et al. (2023b).”

---

> ### Author Response · Authors · 2024-03-17
>
> **W3:** Details of the meta learning approach as well as other aspects of the learning methodology are somewhat limited. It would be helpful to see a more detailed explanation of the method at least in the appendix.
>
> **ANSWER:** We include more details of two fundamental parts of our method, that is the hash encodings and the meta-learning procedure, in the manuscript Appendix A (please refer to the changes in manuscript listed below). For the hash encodings, we now include Figure 4 and description to clarify how the encodings are generated. The figure shows how we determine the hash-encoding embedding for an arbitrary spatial coordinate x. For the meta-learning procedure, we include Figure 5 with a description in the Appendix A that illustrates the learned initialization of the neural fields $f_{ODE}$ and $f_{TAC}$, to support the methods section. The figure shows the output of $f_{TAC}$ after meta-learning for all spatial coordinates sampled at two timepoints. Similarly, it shows the meta-learned $f_{ODE}$ at all spatial coordinates. Both evidently resemble an ‘average’ brain. We also add Figure 6 which shows the progression of $f_{ODE}$ from initialization to the final perfusion representation at various iterations.
>
> **CHANGES IN MANUSCRIPT:**
>
> _Please refer to the updated manuscript for the figures mentioned in the following added paragraphs.._
>
> We add the paragraph _Hash-encodings_ in Appendix A:
>
> “**_Hash-encodings_** _Figure 4 presents an illustrative example of the hash-encoding operation. In the example, we demonstrate hash-encoding with two resolutions ($L=2$) in a two-dimensional setting, while our method actually utilizes $L=16$ resolutions within the three-dimensional domain. The multi-resolution hash-encoding operation divides the domain into multiple grids, with each grid point indexed by an integer. For each grid point, we assign $d=2$ learnable weights in a hash table per resolution, which can be retrieved by looking up the integer index. For a spatial coordinate $\\mathbf{x}$, the hash-encoding operation identifies the weights for the nearest grid points in the hash table for each resolution. Per resolution, the weights corresponding to the grid points are then linearly interpolated according to the relative positions of the coordinate with respect to the grid points. The interpolated weights for each resolution are subsequently stacked to obtain an embedding $e\\in\\mathbb{R}^{Ld}$. This embedding is then used as the input to the networks. We used a PyTorch implementation of multi-resolution hash-encodings (Hsiao, 2023)._”
>
> We add to paragraph _Reptile meta-learning_  in Appendix A:
>
> “_Figure 5 presents the meta-learned initializations for_ $f_{TAC}(\\mathbf{x}, t)$ _at two time points and for_ $f_{ODE}(\\mathbf{x})$ _for the CBF perfusion parameter. We note that the initializations already resemble brain-like attenuation or blood flow values. For instance, there is noticeable increased attenuation in the later timepoint for_ $f_{TAC}(\\mathbf{x}, t)$_’s initialization. Similarly, the initial CBF pattern shows characteristic features, such as increased flow near supplying arteries. Figure 6 presents the output of_ $f_{ODE}(\\mathbf{x})$ _for the CBF at various iterations for ReSPPINN@500 starting from the meta-learned initialization. Moreover, it shows the CBF for ReSPPINN-no-init@5000 for visual comparison. It illustrates_ $f_{ODE}(\\mathbf{x})$_'s iterative progress in achieving a closer fit to the data and accurately determining the perfusion parameters._”

---

> ### Author Response · Authors · 2024-03-17
>
> **Q2:** How is the visual quality of the perfusion image impacted by the hash coding approach? How does the accuracy of the computed perfusion values change without and with the hash coding method?
>
> **ANSWER:**
>
> The hash-encoding plays a fundamental role for the perfusion image quality, allowing for capturing the high frequency details in the data. With identically small networks without hash encoding, fitting the volume-based data would not be possible. Without hash-encoding, even larger networks struggle to accurately fit the data, as seen with SPPINN, which lacks hash encoding but has larger networks for slice-based data. The SPPINN models do not have the capacity to fit volume-based (3D+T) data. Without a good fit, the maps won’t be useful; they might hit a local minimum in terms of residual loss, but the perfusion values won’t be reliable.  Unfortunately, we can not _quantify_ the loss in accuracy, because there's no definitive standard for perfusion values. The closest we can get to evaluating the difference between SPPINN and ReSPPINN is by visually comparing the perfusion maps, or comparing a downstream task such as infarct estimation. Our method clearly offers more detailed perfusion maps.
>
> **CHANGES IN MANUSCRIPT:**
>
> We include additional ReSPPINN qualitative results in the Appendix B Figure 7 for several patients. For comparison, we also include additional SPPINN results in Figure 7 to showcase the effect of the hash-encodings.
>
> Moreover, in Appendix D, we add Table 6 with infarct core segmentation results for slice-based SPPINN perfusion maps, accompanied by the following text:
>
> “_Table 6 lists the infarct core segmentation results using the perfusion maps generated by the original slice-based SPPINN. This experiment resulted in a Dice score of 0.50, an MVD of 10.2 ml, an AVD of 17.3 ml, and an FNR of 0.00. Though the performance is similar to the proposed ReSPPINN approach in terms of Dice score and FNR, the volumetric agreement with the reference DWI segmentation is significantly worse, suggesting an overestimation of the infarct core. This may be a result of a higher level of smoothness in the slice-based perfusion maps, leading to poorly defined segmentation boundaries. Alternatively, the absence of 3D context, and therefore, structural differences between slices, may affect results. It should further be noted that the high difference in computational time (43 minutes for SPPINN, 1.2 minutes for ReSPPINN) makes SPPINN unusable in clinical practice. The slice-wise approach further ignores 3D context, which is supported by our proposed method. For a slice-based approach, fitting perfusion values in the brain's top and bottom parts is challenging and prone to suboptimal optimization, due to the smaller brain area in those regions and the fact that those regions are more prone to image artifacts. A full-volume approach mitigates this by calculating loss across the entire volume, enhancing stability and guiding optimization. The volume-based approach therefore offers significant advantages._”

---

### Official Review · Reviewer_oauj · 2024-02-29

**Confidence:** 3
**Preliminary Rating:** 5
**Recommendation:** Best Paper Award
**Final Rating:** 5

**Summary:**

The paper introduces spatio-temporal perfusion physics-informed neural networks for CT perfusion (CTP) analysis in acute ischemic stroke, leveraging neural fields. This builds upon a previous method, addressing challenges like computation times, 3D vs. 2D-based methods, and the absence of infarct core segmentation to align more closely with clinical practice.

**Strengths:**

- The method employs coordinate-based neural networks or neural fields to estimate perfusion maps. To adapt to 3D, efficient coordinate encodings are utilized, employing hash-encoding to handle high-frequency details.

- Using meta-learning for initialization the authors achieve faster model convergence without compromising performance.

- Good description of the implementation.

- They achieve a computational time that brings the approach closer to clinical practice.

**Weaknesses:**

-  The comparison of the segmentation vs commercial packages that segment based on a threshold is good. Nevertheless, as the authors point-out there are challenges that tackle the segmentation of the core and they achieved a similar performance. Thus, if possible I would add that 'reference' performance in the table, pointing out that it is not in the same data, but to have an idea on what that the actual SOTA is.

**Detailed Comments:**

- In Table 1, report the standard deviation

**Justification Of Final Rating:**

The authors have clarified all my doubts, and my concerns were properly addressed. Furthermore, they did a great effort updating the paper and providing further evidence as well as detailed comments. I think it is a good paper, and I am confident it will be of interest for the community.

**Justification Of The Preliminary Rating:**

The paper builds upon previous work addressing the main challenge of computational time which is a key aspect of an emergency stroke protocol to assess the infarct core area and plan the treatment. Thus, the achieved performance in segmentation, perfusion map reconstruction and computational time all together bring this approach closer to clinical practice. Moreover, the methodology is interesting. Thus, I consider it a very sound and good paper.

**Questions To Address In The Rebuttal:**

- In Section 2.2, the authors point out that the encoding is shared between the $f_{TAC}$ and $f_{ODE}$, but not $f_{AIF}$. Why? Are the coordinates of the f_{AIF} not encoded?

- In the experiment section (3), in the accelerated convergence, the comparison is with $L_{ODE}$ and $L_{TAC}$. Again, why not $L_{AIF}$?

**Special Issue:**

Yes

---

> ### Author Response · Authors · 2024-03-16
>
> We thank the reviewer for their review, helpful comments, and suggestions.
>
> **W1:** The comparison of the segmentation vs commercial packages that segment based on a threshold is good. Nevertheless, as the authors point-out there are challenges that tackle the segmentation of the core and they achieved a similar performance. Thus, if possible I would add that 'reference' performance in the table, pointing out that it is not in the same data, but to have an idea on what that the actual SOTA is.
>
> **ANSWER:** Indeed, adding a 'reference' performance to our manuscript for context is a good suggestion. However, we believe that integrating these results in the same table as ours might lead to confusion due to differences in the underlying data, e.g., different CT perfusion software used for perfusion map generation, larger data set size, and differences in the number of perfusion maps used as input (the ISLES challenge typically involves four perfusion maps and a baseline non-contrast CT, while our study uses only one perfusion map). Accepting the spirit of the suggestion, we include for reference the performance of the top-performing ISLES challenge model in the text, as well as we add an extra Table 7 summarizing the top 5 results from the ISLES challenge in Appendix E.
>
> **CHANGES IN MANUSCRIPT:**
>
> In the Discussion, we add for completeness:
> “Using standard U-Net, our performance is on par with the top methods in the Ischemic Stroke Lesion Segmentation 2018 challenge, _where the leading model achieved a Dice of 0.51_ (Hakim et al., 2021) (see also Appendix E).”
>
> We also add Appendix E and Table 7 with the top five results accompanied by the following text:
> “_Table 7 lists the top-five results of the Ischemic Stroke Lesion Segmentation Challenge (ISLES) 2018 (Hakim et al., 2021) as detailed in the official leaderboard at <https://www.smir.ch/ISLES/Start2018>. The goal of the challenge was to segment the acute phase DWI MRI reference infarct core from CT and CTP imaging. The available data (63 patients for training, 40 patients for testing) comprised baseline non-contrast CT, CTP source data, and four perfusion maps (CBF, CBV, MTT, Tmax) generated by RAPID (RAPID; iSchemaview, Menlo Park, CA). The reference standard was segmented on acute phase DWI MRI with a median time between CT and MRI of 36 minutes. Most participants of the challenge employed the non-contrast CT and four perfusion maps (CBF, CBV, MTT, Tmax) as inputs to the segmentation models. We refer to Hakim et al., (2021) for a full overview of the challenge and main results. The Dice score of ReSPPINN is competitive, aligning with the top-five methods featured on the ISLES 2018 leaderboard. Notably, for ReSPPINN, the absolute volumetric difference is approximately 3-5 ml higher._”
>
> **DC1:** In Table 1, report the standard deviation
>
> **ANSWER and CHANGES IN MANUSCRIPT:** We now include the standard deviation over 5 runs with different seeds in Table 1.
>
> We add a new caption to Table 1:
>
> “Dice and mean or absolute volumetric difference (MVD, AVD), _and false negative rate (FNR). We report mean (standard deviation) for 5 seeds._”
>
> **Q1:** In Section 2.2, the authors point out that the encoding is shared between the $f_{TAC}$ and $f_{ODE}$, but not $f_{AIF}$. Why? Are the coordinates of the $f_{AIF}$ not encoded?
>
> **ANSWER:** We only apply the hash encoding to the spatial coordinates in $f_{TAC}$ and $f_{ODE}$, because in contrast to the temporal domain, we expect high frequency variation in the spatial domain. For the $f_{AIF}$ we do not require encoding, since the 1 dimensional AIF is very simple to approximate and already efficient without requiring hashing. Moreover, sharing these encoding with $f_{AIF}$ is not possible because that neural field does not have spatial inputs.
>
> **CHANGES IN MANUSCRIPT:**
>
> Following your remark, we emphasize this in the method section:
>
> _“We share the encoding layer between_ $f_{ODE}(\mathbf{x})$ _and_ $f_{TAC}(\mathbf{x},t)$. _For_ $f_{AIF}(t)$ _we do not require encoding, since approximating the one-dimensional AIF is already efficient without hashing._”
>
> **Q2:** In the experiment section (3), in the accelerated convergence, the comparison is with $L_{ODE}$ and $L_{TAC}$ . Again, why not $L_{AIF}$?
>
> **ANSWER:** We have not included $L_{AIF}$ in our analysis because fitting the AIF data with a small neural network is straightforward, with the loss quickly decreasing after just a few iterations to achieve an almost perfect fit, irrespective of initialization. Hence, $L_{AIF}$  does not provide many additional insights regarding model performance.
>
> **CHANGES IN MANUSCRIPT:**
>
> We add in the experiments section:
>
> “_We exclude_ $L_{AIF}$ _from evaluation as_ $f_{AIF}(t)$ _fits the AIF data fast regardless of initialization._”

---

### Meta-Review · Area_Chair_RVFe · 2024-04-04

**Recommendation:** Accept (Oral)
**Confidence:** 5

**Metareview:**

The authors answered the reviewers' questions with great attention to detail, emphasizing their dedication and the importance of their work. In addition, they have considerably improved an already outstanding manuscript, introducing new perspectives that strengthen the results. I am convinced that this submission will be very attractive to the MIDL community, and I therefore recommend accepting the article for publication in MIDL.

---

### Decision · Program_Chairs · 2024-04-06

Accept (Oral)